# Non-equilibrium evolution of Bose-Einstein condensate deformation in temporally controlled weak disorder

**Milan Radonjić[1,2⋆] and Axel Pelster[1]**

**1** Physics Department and Research Center OPTIMAS, Technical University of Kaiserslautern, Erwin-Schrödinger Straße 46, 67663 Kaiserslautern, Germany
**2** Institute of Physics Belgrade, University of Belgrade, Pregrevica 118, 11080 Belgrade, Serbia

⋆ milanr@ipb.ac.rs

## Abstract

We consider a time-dependent extension of a perturbative mean-field approach to the homogeneous dirty boson problem by considering how switching on and off a weak disorder potential affects the stationary state of an initially equilibrated Bose-Einstein condensate by the emergence of a disorder-induced condensate deformation. We find that in the switch on scenario the stationary condensate deformation turns out to be a sum of an equilibrium part, that actually corresponds to adiabatic switching on the disorder, and a dynamically-induced part, where the latter depends on the particular driving protocol. If the disorder is switched off afterwards, the resulting condensate deformation acquires an additional dynamically-induced part in the long-time limit, while the equilibrium part vanishes. We also present an appropriate generalization to inhomogeneous trapped condensates. Our results demonstrate that the condensate deformation represents an indicator of the generically non-equilibrium nature of steady states of a Bose gas in a temporally controlled weak disorder.

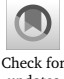
# 1 Introduction

All realistic physical systems inevitably involve a certain level of disorder due to the generic presence of an environment or a random distribution of imperfections. This is commonly understood as a nuisance which has to be coped with, especially in solid-state materials. However, a complementary and a more nuanced viewpoint is recently gaining the ground [1]. Namely, the disorder can lead to exciting, qualitatively novel phenomena that do not have any clean counterparts, such as the well investigated phenomenon of Anderson localization [2,3]. Thus, one has a potential of utilizing the disorder in order to engineer new states of matter, where the many-body localized phase is a recent prominent example [4,5], which is currently being debated [6]. This is in line with an ongoing quest to exploit the system environment as a tunable knob for quantum control [7–13].

Seminal studies of superfluid helium in porous vycor glass [14, 15] instigated a theoretical interest in the so-called dirty boson problem that deals with the emergence of different phases of ultracold bosonic atoms in the presence of a static random potential. Starting with the experimental realization of Bose-Einstein condensates (BECs) in 1995 [16, 17], there was a renewed theoretical effort in understanding the influence of disorder on samples of cold atoms [18]. Huang and Meng made significant progress in describing the thermodynamic equilibrium by developing a Bogoliubov treatment of a weakly interacting homogeneous Bose gas in a static delta-correlated random potential, where quantum, thermal and disorder fluctuations were assumed to be small [19]. After accounting for the disorder perturbatively they found that superfluidity persists in spite of it, although the superfluid depletion turned out to be larger than the condensate depletion. Their work was subsequently expanded by many others, using the original approach of second quantization [20–23] or employing the functional integral framework and the replica method [24–26]. The consideration of scenarios when the disorder correlation function has a non-zero correlation length $\sigma$ showed that both condensate and superfluid depletions decrease with increasing $\sigma$, in particular for the case of a Gaussian [21,27–29], Lorentzian [30], or laser speckle disorder [31,32]. Quite recently, the predictions of the Huang-Meng theory were experimentally confirmed by analyzing the cloud shape of a molecular $^6$Li BEC in a disordered trap [33]. Furthermore, building on an equilibrium inhomogeneous Bogoliubov theory [34], the results presented in [35] clarified the relation between the condensate depletion and condensate deformation in the presence of a weak, possibly random, external potential. We adopt such a notion of the condensate deformation here, use it as a measure of the average effect of the disorder and generalize it to the non-equilibrium context.

Quench dynamics is an active field of research encompassing condensed matter physics and quantum information, with many intriguing applications to ultracold atomic gases [36]. In a recent experiment [37] quantum quenches of disorder in an ultracold bosonic gas in a lattice were used to dynamically probe the superfluid-Bose glass quantum phase transition at non-zero temperature. It is well known that quenches may lead to the appearance of nontrivial steady states. For instance, in our context, the authors of [38] studied an interaction quench of a 3D BEC in a static disordered potential and concluded that in the long-time limit the superfluid density tends to zero while the condensate density remains finite, i.e., that the quench dynamics enhances the ability of disorder to deplete the superfluid more than to deform the condensate. However, a quench is just a limiting case of a more general switch on and off protocol, which is a ubiquitous basic element of any experiment. In this work we consider a more generic scenario of switching the disorder on (off) during a certain time interval, while keeping the interaction strength constant, and we investigate the resulting dynamical behavior of the condensate deformation. Here we are, in particular, interested in the long-time limit of the emerging condensate deformation and in its tunability upon changing the respective

system parameters.

To this end, the paper is organized as follows. In Sec. 2 we present a general mean-field theory for calculating the condensate deformation of a homogeneous Bose gas in a temporally controlled weak disorder. The relevant case studies involving the disorder switch on (off) scenarios are worked out and discussed in Sec. 3. The extension to inhomogeneous condensates in disordered traps is given in Sec. 4, followed by concluding remarks in Sec. 5.

## 2   General theory

We consider $N$ identical weakly interacting ultracold bosons in a 3D box of large volume $V$, under the appropriate periodic boundary conditions. The weakly interacting theory of dirty bosons [19–22] demonstrates that Bogoliubov quasiparticles and disorder-induced fluctuations decouple in the lowest order. Thus, the leading correction due to the presence of a disorder potential is derivable from a mean-field theory [28, 30]. Therefore, we suppose that at time $t = 0$ the system is in its equilibrium ground state, which is described by the macroscopic wave function $\Psi_0(\mathbf{x})$ that solves the stationary Gross-Pitaevskii equation

$$\left(-\frac{\hbar^2 \nabla^2}{2m} - \mu_0 + g|\Psi_0(\mathbf{x})|^2\right)\Psi_0(\mathbf{x}) = 0. \tag{1}$$

The particle density $n = N/V = |\Psi_0(\mathbf{x})|^2$ determines the equilibrium chemical potential $\mu_0 = gn$. In the following we take, without loss of generality, a real-valued clean-case homogeneous wave function $\Psi_0(\mathbf{x}) = \sqrt{n}$. The thermodynamic limit $N \to \infty$, $V \to \infty$ with $n = const$ will be implicitly assumed at the final stage of all calculations.

At times $t \geq 0$ an external disorder potential is switched on

$$u(\mathbf{x}, t) = u(\mathbf{x})f(t), \tag{2}$$

where $u(\mathbf{x})$ is a random potential with the corresponding ensemble average $\langle \ldots \rangle$, and $f(t)$ is a deterministic driving function such that $f(0) = 0$ and $0 \leq f(t) \leq 1$. We assume that the disorder potential has zero ensemble average at every point, $\langle u(\mathbf{x}) \rangle = 0$, in order to eliminate the effects of a simple shift of the chemical potential. The two-point correlation function is supposed to be of the form

$$\langle u(\mathbf{x})u(\mathbf{x}') \rangle = \mathcal{R}(\mathbf{x} - \mathbf{x}'), \tag{3}$$

so that homogeneity is restored after performing the spatial ensemble average. The average height/depth of the hills/valleys of the random potential landscape is related to $\mathcal{R}(\mathbf{0})$. The disorder correlation length $\sigma$ measures the average width of these hills/valleys, and $\mathcal{R}(\mathbf{x} - \mathbf{x}')$ typically decays down to zero for distances $|\mathbf{x} - \mathbf{x}'|$ larger than several $\sigma$. In the $\mathbf{k}$-space we have, correspondingly,

$$\langle u(\mathbf{k}) \rangle = 0, \quad \langle u(\mathbf{k})u(\mathbf{k}') \rangle = (2\pi)^3 \delta(\mathbf{k} + \mathbf{k}')\mathcal{R}(\mathbf{k}). \tag{4}$$

The disorder strength, defined as $\mathcal{R}(\mathbf{k} = \mathbf{0})$, takes simultaneously into account all the spatial properties of the random potential.

The system dynamics at $t \geq 0$ is described by the time-dependent Gross-Pitaevskii equation, which reads after explicitly separating out the initial phase evolution $e^{-i\mu_0 t/\hbar}$,

$$i\hbar \frac{\partial}{\partial t}\Psi(\mathbf{x}, t) = \left(-\frac{\hbar^2 \nabla^2}{2m} + u(\mathbf{x})f(t) - \mu_0 + g|\Psi(\mathbf{x}, t)|^2\right)\Psi(\mathbf{x}, t). \tag{5}$$

We work in the regime where $u(\mathbf{x})$ is small in comparison with all other energy scales of the system, and therefore we can treat its influence in a perturbative manner. Later we state a quantitative condition that should be satisfied by weak disorder. The random potential slightly deforms the condensate and we make a perturbative ansatz for the wave function

$$\Psi(\mathbf{x}, t) = \Psi_0(\mathbf{x}) + \Psi_1(\mathbf{x}, t) + \Psi_2(\mathbf{x}, t) + \dots, \tag{6}$$

where $|\Psi_\alpha(\mathbf{x}, t)| = \mathcal{O}(|u(\mathbf{x})|^\alpha)$ denote perturbative corrections due to the disorder and obey the initial condition $\Psi_\alpha(\mathbf{x}, 0) = 0$ for all $\alpha \geq 1$. Following Ref. [39], the difference between the disorder-averaged particle density $\langle |\Psi(\mathbf{x}, t)|^2 \rangle$ and the density of the disorder-averaged condensate $|\langle \Psi(\mathbf{x}, t) \rangle|^2$,

$$q(t) = \langle |\Psi(\mathbf{x}, t)|^2 \rangle - |\langle \Psi(\mathbf{x}, t) \rangle|^2 \equiv \langle |\Psi(\mathbf{x}, t) - \langle \Psi(\mathbf{x}, t) \rangle|^2 \rangle, \tag{7}$$

stands out as a possible dynamical extension of the Bose-glass order parameter [38], in close analogy to the well-known Edwards-Anderson order parameter for spin glasses [40]. The last expression in (7) enables us to interpret it as the average particle density associated with disorder-induced condensate fluctuations. In agreement with the in-depth discussion of [35], it represents the ensemble average of the condensate deformation due to the disorder. Henceforth, we simply call it the condensate deformation. As we are going to show, its value even represents an indicator for the non-equilibrium feature of the system's stationary states.

Using the perturbative expansion (6), we obtain for the average particle density

$$\begin{aligned}
\langle |\Psi(\mathbf{x}, t)|^2 \rangle = \Psi_0^2 &+ \Psi_0 \big( \langle \Psi_1(\mathbf{x}, t) \rangle + \langle \Psi_1^*(\mathbf{x}, t) \rangle \big) \\
&+ \big[ \langle |\Psi_1(\mathbf{x}, t)|^2 \rangle + \Psi_0 \big( \langle \Psi_2(\mathbf{x}, t) \rangle + \langle \Psi_2^*(\mathbf{x}, t) \rangle \big) \big] + \dots,
\end{aligned} \tag{8}$$

where the terms are grouped according to their respective perturbative order. On the other hand, the density of the disorder-averaged condensate is

$$\begin{aligned}
|\langle \Psi(\mathbf{x}, t) \rangle|^2 = \Psi_0^2 &+ \Psi_0 \big( \langle \Psi_1(\mathbf{x}, t) \rangle + \langle \Psi_1^*(\mathbf{x}, t) \rangle \big) \\
&+ \big[ |\langle \Psi_1(\mathbf{x}, t) \rangle|^2 + \Psi_0 \big( \langle \Psi_2(\mathbf{x}, t) \rangle + \langle \Psi_2^*(\mathbf{x}, t) \rangle \big) \big] + \dots,
\end{aligned} \tag{9}$$

so that, up to the second order, the condensate deformation becomes

$$q(t) = \langle |\Psi_1(\mathbf{x}, t)|^2 \rangle - |\langle \Psi_1(\mathbf{x}, t) \rangle|^2 + \dots. \tag{10}$$

Therefore, for calculations to that order, we only need the first perturbative correction $\Psi_1(\mathbf{x}, t)$. The first-order coupled equations follow from (5) and read

$$i\hbar \frac{\partial}{\partial t} \Psi_1(\mathbf{x}, t) = \left( -\frac{\hbar^2 \nabla^2}{2m} + g\Psi_0^2 \right) \Psi_1(\mathbf{x}, t) + g\Psi_0^2 \Psi_1^*(\mathbf{x}, t) + \Psi_0 u(\mathbf{x}) f(t), \tag{11a}$$

$$-i\hbar \frac{\partial}{\partial t} \Psi_1^*(\mathbf{x}, t) = \left( -\frac{\hbar^2 \nabla^2}{2m} + g\Psi_0^2 \right) \Psi_1^*(\mathbf{x}, t) + g\Psi_0^2 \Psi_1(\mathbf{x}, t) + \Psi_0 u(\mathbf{x}) f(t). \tag{11b}$$

With the help of the Fourier transform and its inverse

$$u(\mathbf{k}) = \int_{\mathbb{R}^3} d^3x \, e^{-i\mathbf{k}\mathbf{x}} u(\mathbf{x}), \quad u(\mathbf{x}) = \int_{\mathbb{R}^3} \frac{d^3k}{(2\pi)^3} e^{i\mathbf{k}\mathbf{x}} u(\mathbf{k}), \tag{12}$$

we get

$$i\hbar \frac{\partial}{\partial t} \Psi_1(\mathbf{k}, t) = \big( \hbar\omega_{\mathbf{k}} + g\Psi_0^2 \big) \Psi_1(\mathbf{k}, t) + g\Psi_0^2 \Psi_1^*(\mathbf{k}, t) + \Psi_0 u(\mathbf{k}) f(t), \tag{13a}$$

$$-i\hbar\frac{\partial}{\partial t}\Psi_1^*(\mathbf{k},t) = \left(\hbar\omega_\mathbf{k} + g\Psi_0^2\right)\Psi_1^*(\mathbf{k},t) + g\Psi_0^2\Psi_1(\mathbf{k},t) + \Psi_0 u(\mathbf{k})f(t), \tag{13b}$$

where $\hbar\omega_\mathbf{k} = \hbar^2\mathbf{k}^2/(2m)$ is the free particle dispersion. In order to automatically incorporate the initial conditions $\Psi_1^{(*)}(\mathbf{k},t=0) = 0$, we apply the Laplace transform

$$\mathcal{L}[f](s) = \int_0^\infty dt\, f(t)e^{-st}, \tag{14}$$

and make the identification $\mathcal{L}[f](s) \equiv f(s)$, for brevity of notation. With this, the differential equations (13) reduce to a system of algebraic relations

$$\left(\hbar\omega_\mathbf{k} + g\Psi_0^2 - i\hbar s\right)\Psi_1(\mathbf{k},s) + g\Psi_0^2\Psi_1^*(\mathbf{k},s) = -\Psi_0 u(\mathbf{k})f(s), \tag{15a}$$

$$g\Psi_0^2\Psi_1(\mathbf{k},s) + \left(\hbar\omega_\mathbf{k} + g\Psi_0^2 + i\hbar s\right)\Psi_1^*(\mathbf{k},s) = -\Psi_0 u(\mathbf{k})f(s), \tag{15b}$$

which are solved by

$$\Psi_1(\mathbf{k},s) = -\frac{\Psi_0}{\hbar}\frac{\omega_\mathbf{k} + is}{\Omega_\mathbf{k}^2 + s^2}u(\mathbf{k})f(s), \tag{16a}$$

$$\Psi_1^*(\mathbf{k},s) = -\frac{\Psi_0}{\hbar}\frac{\omega_\mathbf{k} - is}{\Omega_\mathbf{k}^2 + s^2}u(\mathbf{k})f(s), \tag{16b}$$

with the Bogoliubov dispersion $\hbar\Omega_\mathbf{k} = \sqrt{\hbar\omega_\mathbf{k}\left(\hbar\omega_\mathbf{k} + 2g\Psi_0^2\right)}$. Using the inverse Laplace transform

$$\frac{1}{\hbar}\frac{\omega_\mathbf{k} + is}{\Omega_\mathbf{k}^2 + s^2} \xrightarrow{\mathcal{L}^{-1}} \frac{1}{\hbar}\left\{\frac{\omega_\mathbf{k}}{\Omega_\mathbf{k}}\sin(\Omega_\mathbf{k} t) + i\cos(\Omega_\mathbf{k} t)\right\} \equiv \mathcal{K}(\mathbf{k},t), \tag{17}$$

we finally find

$$\Psi_1(\mathbf{k},t) = -\Psi_0 u(\mathbf{k})\int_0^t dt'\,\mathcal{K}(\mathbf{k},t-t')f(t'), \tag{18a}$$

$$\Psi_1^*(\mathbf{k},t) = -\Psi_0 u(\mathbf{k})\int_0^t dt'\,\mathcal{K}^*(\mathbf{k},t-t')f(t'). \tag{18b}$$

We note that, since $\langle u(\mathbf{k})\rangle = 0$, it follows $\langle\Psi_1^{(*)}(\mathbf{k},t)\rangle = 0$ and consequently $\langle\Psi_1^{(*)}(\mathbf{x},t)\rangle = 0$. The above expressions immediately provide us with $\Psi_1^{(*)}(\mathbf{x},t)$, reducing the condensate deformation (10) to the expression

$$q(t) = n\int_{\mathbb{R}^3}\frac{d^3k}{(2\pi)^3}\mathcal{R}(\mathbf{k})\left|\int_0^t dt'\,\mathcal{K}(\mathbf{k},t-t')f(t')\right|^2, \tag{19}$$

where we used (4) and $\mathcal{K}^{(*)}(-\mathbf{k},t-t') = \mathcal{K}^{(*)}(\mathbf{k},t-t')$. In the next section we demonstrate that the last expression represents a time-dependent generalization of the seminal Huang and Meng equilibrium result [19].

## 3   Switch on – switch off case studies

In this section we apply the developed theory to two particular relevant scenarios. First, we focus our attention on an exponential switching on of the disorder potential, which leads to two physically distinct contributions to the stationary condensate deformation, one of which is generically of a non-equilibrium nature. Afterwards, we analyze the fate of both contributions upon switching the disorder off.

## 3.1 Switch on scenario

Let us first consider the case of an exponential introduction of the disorder potential via

$$f(t) = 1 - e^{-t/\tau}, \tag{20}$$

where $\tau > 0$ determines the characteristic time scale. In this way one has $f(0) = 0$ and stationarity occurs for $t \gg \tau$ since $f(t) \to 1$ at large times. Hence, from (17) and (19) we obtain

$$q_\tau(t) = n \int_{\mathbb{R}^3} \frac{d^3k}{(2\pi)^3} \mathcal{R}(\mathbf{k}) \left\{ \frac{\omega_\mathbf{k}^2}{\hbar^2 \Omega_\mathbf{k}^4} + \frac{\omega_\mathbf{k}^2 + \Omega_\mathbf{k}^2}{2\hbar^2 \Omega_\mathbf{k}^4 (1 + \tau^2 \Omega_\mathbf{k}^2)} - \frac{2\omega_\mathbf{k}^2 \cos(\Omega_\mathbf{k} t)}{\hbar^2 \Omega_\mathbf{k}^4 (1 + \tau^2 \Omega_\mathbf{k}^2)} - \frac{2\tau \omega_\mathbf{k}^2 \sin(\Omega_\mathbf{k} t)}{\hbar^2 \Omega_\mathbf{k}^3 (1 + \tau^2 \Omega_\mathbf{k}^2)} \right.$$
$$+ \frac{(\Omega_\mathbf{k}^2 - \omega_\mathbf{k}^2)(\tau^2 \Omega_\mathbf{k}^2 - 1) \cos(2\Omega_\mathbf{k} t)}{2\hbar^2 \Omega_\mathbf{k}^4 (1 + \tau^2 \Omega_\mathbf{k}^2)^2} - \frac{\tau(\Omega_\mathbf{k}^2 - \omega_\mathbf{k}^2) \sin(2\Omega_\mathbf{k} t)}{\hbar^2 \Omega_\mathbf{k}^3 (1 + \tau^2 \Omega_\mathbf{k}^2)^2} + e^{-2t/\tau} \frac{\tau^2 (1 + \tau^2 \omega_\mathbf{k}^2)}{\hbar^2 (1 + \tau^2 \Omega_\mathbf{k}^2)^2}$$
$$+ e^{-t/\tau} \left[ \frac{2\tau^2 (\omega_\mathbf{k}^2 - \Omega_\mathbf{k}^2) \cos(\Omega_\mathbf{k} t)}{\hbar^2 \Omega_\mathbf{k}^2 (1 + \tau^2 \Omega_\mathbf{k}^2)^2} - \frac{2\tau^2 \omega_\mathbf{k}^2}{\hbar^2 \Omega_\mathbf{k}^2 (1 + \tau^2 \Omega_\mathbf{k}^2)} + \frac{2\tau(1 + \tau^2 \omega_\mathbf{k}^2) \sin(\Omega_\mathbf{k} t)}{\hbar^2 \Omega_\mathbf{k} (1 + \tau^2 \Omega_\mathbf{k}^2)^2} \right] \right\}. \tag{21}$$

In the long-time limit we can safely neglect the exponentially decaying terms. In addition, the 3D $\mathbf{k}$-space integral of the remaining terms, that involve trigonometric functions, effectively represents a sum of infinitely many rapidly oscillating functions having incommensurate periods, which turns out to vanish as $t \to \infty$. This leads to the stationary condensate deformation

$$q_\tau \equiv \lim_{t \to \infty} q_\tau(t) = n \int_{\mathbb{R}^3} \frac{d^3k}{(2\pi)^3} \mathcal{R}(\mathbf{k}) \left\{ \frac{\omega_\mathbf{k}^2}{\hbar^2 \Omega_\mathbf{k}^4} + \frac{\omega_\mathbf{k}^2 + \Omega_\mathbf{k}^2}{2\hbar^2 \Omega_\mathbf{k}^4 (1 + \tau^2 \Omega_\mathbf{k}^2)} \right\}, \tag{22}$$

which notably consists of two physically different terms. The first one is the same for any switch on time and, thus, is not of dynamical origin. In fact, in the case of an adiabatic switching on of the disorder potential, which corresponds to the limit $\tau \to \infty$, only that term remains, so that we find

$$\lim_{\tau \to \infty} q_\tau = n \int_{\mathbb{R}^3} \frac{d^3k}{(2\pi)^3} \mathcal{R}(\mathbf{k}) \frac{\omega_\mathbf{k}^2}{\hbar^2 \Omega_\mathbf{k}^4} = n \int_{\mathbb{R}^3} \frac{d^3k}{(2\pi)^3} \frac{\mathcal{R}(\mathbf{k})}{(\hbar \omega_\mathbf{k} + 2gn)^2}. \tag{23}$$

This is exactly the part of the condensate deformation that corresponds to the equilibrium value [23, 28, 30, 41]. The presence of an excess deformation in (22) signals the non-equilibrium

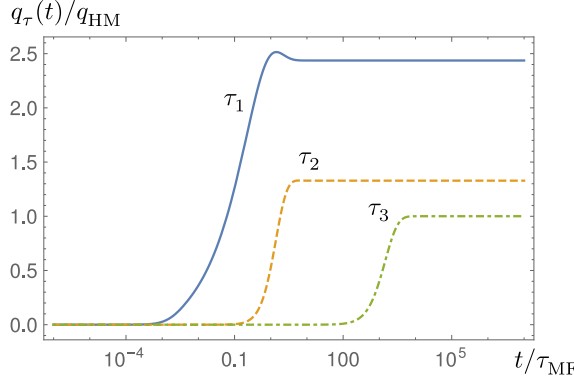

Figure 1: Temporal evolution of condensate deformation (21) in units of the equilibrium deformation (26) for exponential temporal increase (20) of delta-correlated disorder potential (25), with the time scales $10^3 \tau_1 = \tau_2 = 10^{-3} \tau_3 = \tau_{\mathrm{MF}} \equiv \hbar/(gn)$.

feature of the achieved stationary state in general. Therefore, we arrive at a straightforward interpretation of the total stationary condensate deformation $q_\tau$ in (22): the first term is the equilibrium part (23), while the second one is dynamically induced and depends on the switch on time scale $\tau$ and, thus, on the respective driving protocol. The dynamically induced contribution clearly ranges from zero in the adiabatic case up to the maximal value attained for a sudden quench of the disorder potential, which corresponds to the limit $\tau \to 0$. In the latter case, we have

$$\lim_{\tau \to 0} q_\tau = n \int_{\mathbb{R}^3} \frac{d^3 k}{(2\pi)^3} \mathcal{R}(\mathbf{k}) \frac{3\omega_{\mathbf{k}}^2 + \Omega_{\mathbf{k}}^2}{2\hbar^2 \Omega_{\mathbf{k}}^4} = n \int_{\mathbb{R}^3} \frac{d^3 k}{(2\pi)^3} \frac{\mathcal{R}(\mathbf{k})(2\hbar\omega_{\mathbf{k}} + g n)}{\hbar\omega_{\mathbf{k}}(\hbar\omega_{\mathbf{k}} + 2g n)^2} . \tag{24}$$

Note that there are some constraints concerning the switch on time $\tau$ in the experimentally feasible cold-atom setups. Due to technical limitations how fast the laser power can be controlled, $\tau$ cannot go much below 0.1 $\mu s$. On the other hand, three-body losses and collisions with hot particles from the background pressure in the vacuum chamber lead to a finite lifetime of the condensate, which is of the order of 10 s. Hence, realizing $\tau \gtrsim 1$ s might be problematic due to these experimental limiting factors.

The above findings, which are applicable to any disorder correlation $\mathcal{R}(\mathbf{x})$, will now be illustrated for a relevant special choice of the spatially delta-correlated disorder

$$\mathcal{R}(\mathbf{x}) = R\, \delta(\mathbf{x} - \mathbf{x}'), \qquad \mathcal{R}(\mathbf{k}) = R . \tag{25}$$

Any effect of such a disorder represents the upper bound for every other correlated disorder scenario, because the influence of the disorder tends to decrease with increasing its correlation length. Such a disorder can even be realized experimentally via a random distribution of many neutral atomic impurities trapped in a deep optical lattice [42–44].

From the Huang-Meng theory [19] we know that the equilibrium condensate deformation (23) for the delta-correlated disorder (25) reads

$$q_{\mathrm{HM}} = R \frac{m^{3/2}}{4\pi\hbar^3 \sqrt{g}} \sqrt{n} . \tag{26}$$

The general time-dependent result for the condensate deformation (21) is presented in Fig. 1 for three switch on times $\tau$. The full blue curve corresponds to a rapid quench, i.e. for $\tau \to 0$. Following an initial increase the condensate deformation displays overshooting and eventually settles in a new stationary state. The green dot-dashed curve depicts the approach to the

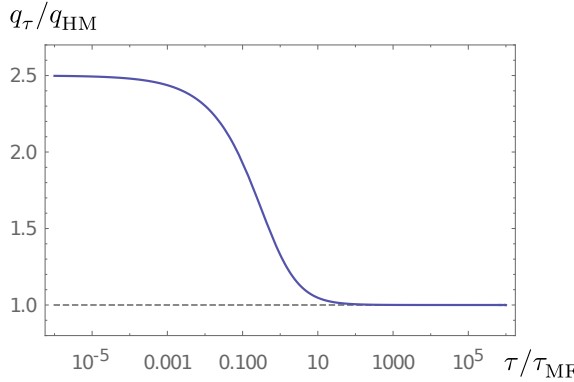

Figure 2: Stationary condensate deformation (27) versus the disorder switch on time scale $\tau$. The equilibrium deformation (26) is reached after a slow adiabatic introduction of a delta-correlated disorder potential via (20).

steady state in the opposite adiabatic regime, i.e. for $\tau \to \infty$. Here, the overshooting of the condensate deformation is absent for a slow enough switch on driving. Note that the stationary long-time limit can even be determined analytically from (22) using (25) for arbitrary time scale $\tau$, and is given by

$$q_\tau = F(\tau/\tau_{\mathrm{MF}})q_{\mathrm{HM}}, \tag{27}$$

where $\tau_{\mathrm{MF}} \equiv \hbar/(gn)$ represents the characteristic mean-field time scale and we introduced the form-function

$$F(\lambda) = 5/2 - 4\lambda^2 + 4(\lambda - 1/2)\sqrt{\lambda(\lambda+1)}, \tag{28}$$

which is plotted in Fig. 2. From (27) and (28) we straight-forwardly deduce the special cases

$$\lim_{\tau\to\infty} q_\tau = q_{\mathrm{HM}}, \qquad \lim_{\tau\to 0} q_\tau = \frac{5}{2}q_{\mathrm{HM}}. \tag{29}$$

We again see that, after the adiabatic introduction of the disorder potential, the new deformed stationary state is reached, which corresponds precisely to the equilibrium result (26) [19]. Moreover, after a sudden quench of the disorder the system does not equilibrate at all, since the stationary condensate deformation is $5/2$ times larger than the equilibrium one. The additional contribution of $3q_{\mathrm{HM}}/2$ comes from the sudden quench itself and it represents the maximally possible dynamically generated condensate deformation for weak delta-correlated disorder.

## 3.2 Switch on – switch off scenario

Having identified the equilibrium and the purely dynamical contribution to the condensate deformation, we now aim at examining their fate upon switching the disorder off. Thus, we next focus on the following switch on − switch off scenario, which is given by the driving function

$$f(t) = \begin{cases} \left[1 - \exp\left(-\dfrac{t}{\tau_1}\right)\right]\left[1 - \exp\left(\dfrac{t-2\tau}{\tau_2}\right)\right], & 0 \le t \le 2\tau, \\ 0, & \text{otherwise,} \end{cases} \tag{30}$$

and is depicted in Fig. 3. It corresponds to an exponential switching on with the characteristic time $\tau_1$, followed by a switch off with the decrease time $\tau_2$. The disorder disappears at $t = 2\tau$. The time $\tau$ has to be much longer than $\tau_1, \tau_2$ so that about $t \gtrsim \tau$ a transient stationary

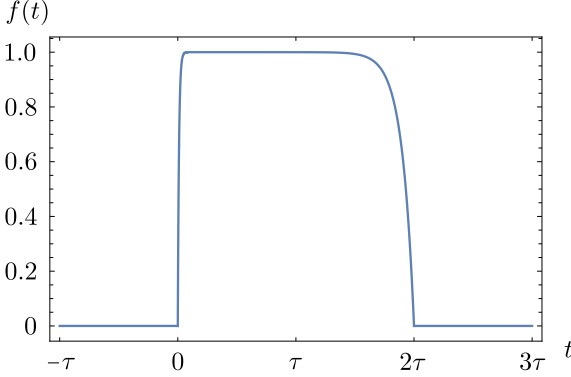

Figure 3: Exemplary switch on – switch off driving function (30) for $\tau = 100\tau_1 = 10\tau_2$.

Table 1: Stationary condensate deformation (31) for delta-correlated disorder (25) switched on and off via (30), either adiabatically or through a rapid quench.

| $q_{\tau_1,\tau_2}$ | $\tau_2 \to \infty$ | $\tau_2 \to 0$ |
|---|---|---|
| $\tau_1 \to \infty$ | 0 | $\dfrac{3}{2}q_{\text{HM}}$ |
| $\tau_1 \to 0$ | $\dfrac{3}{2}q_{\text{HM}}$ | $3q_{\text{HM}}$ |

state, like in the previous scenario, is effectively reached. Hence, $\tau$ approximately determines the disorder duration time. For $t \gg 2\tau$ a final stationary state is achieved. Using similar methodology as before, we find the final state condensate deformation

$$q_{\tau_1,\tau_2} \equiv \lim_{t\to\infty} q_{\tau_1,\tau_2}(t) = n \int_{\mathbb{R}^3} \frac{d^3k}{(2\pi)^3} \mathcal{R}(\mathbf{k}) \left[ \frac{\omega_{\mathbf{k}}^2 + \Omega_{\mathbf{k}}^2}{2\hbar^2 \Omega_{\mathbf{k}}^4 (1 + \tau_1^2 \Omega_{\mathbf{k}}^2)} + \frac{\omega_{\mathbf{k}}^2 + \Omega_{\mathbf{k}}^2}{2\hbar^2 \Omega_{\mathbf{k}}^4 (1 + \tau_2^2 \Omega_{\mathbf{k}}^2)} \right]. \quad (31)$$

First of all, we observe that the equilibrium contribution to the condensate deformation vanishes, as expected, once the disorder has finally disappeared. Furthermore, the resulting condensate deformation is dynamically accumulated during both the switching on and the switching off parts of the driving protocol. In the considered case, the dependence on the respective characteristic time scales $\tau_1, \tau_2$ is identical for both parts. Hence, the time-reversed protocol would have led to the same end result. With this we conclude that the equilibrium condensate deformation is the only part that can be remedied in a reversible fashion in the switch on – switch off scenario. The previous finding is illustrated in Tab. 1 for the example of the delta-correlated disorder. Thus, the adiabatic switch on followed by the sudden quench down and the sudden quench up followed by the adiabatic switch off scenarios lead both to the same final condensate deformation of $3q_{\text{HM}}/2$. Furthermore, the largest dynamically generated deformation is three times larger than the equilibrium one (26) and it occurs by suddenly switching the disorder on and off.

### 3.3 Finite correlation length case

Finally, we analyze the effect of a finite correlation length $\sigma$ on the condensate deformation. For this purpose we choose a Gaussian correlation function of the disorder,

$$\mathcal{R}(\mathbf{x}) = R \frac{e^{-\mathbf{x}^2/\sigma^2}}{\pi^{3/2}\sigma^3}, \qquad \mathcal{R}(\mathbf{k}) = R\, e^{-\sigma^2 \mathbf{k}^2/4}, \quad (32)$$

which has the same strength $\mathcal{R}(\mathbf{k}=\mathbf{0}) = R$ as in the delta-correlated case. Experimental realization of such a disorder can be achieved by using laser speckles [33, 45, 46]. Since the dynamical effects are revealed already during switching on, we simply consider the driving function (20). The resulting condensate deformation (22) is plotted in Fig. 4 for a range of switch on times $\tau$ and correlation lengths $\sigma$. As expected, for $\sigma \ll \xi$, where $\xi = \hbar/\sqrt{2mgn}$ denotes the condensate healing length, the behavior is approaching the delta-correlated case discussed in subsection 3.1. The increase of the correlation length $\sigma$ leads to a drop of the stationary condensate deformation $q_\tau$ below $q_{\text{HM}}$ and further towards zero for $\sigma \gg \xi$, even for very rapid quenches of the disorder. This is in accordance with the general wisdom that the disorder influence decreases when $\sigma$ becomes larger.

Let us briefly comment on when a disorder potential can be considered as "weak". To this end, it is necessary that the equilibrium condensate deformation (23) is much smaller than

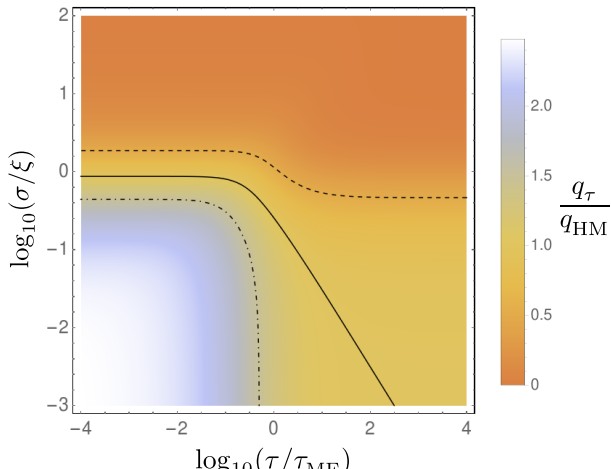

Figure 4: Dependence of the stationary condensate deformation (22) on the time scale $\tau$ of exponential switch on (20) of the Gaussian-correlated disorder potential (32) and on its correlation length $\sigma$. The dot-dashed, full, and dashed curves correspond to $q_\tau/q_{\mathrm{HM}}$ being equal to 2, 1, and 1/2, respectively.

the particle density. The sufficient condition for it is that the Huang-Meng deformation (26) for the delta-correlated disorder of the same strength $R = \mathcal{R}(\mathbf{k} = \mathbf{0})$ is much smaller than $n$. This places the following requirement on the disorder strength

$$\frac{2^{3/2}\pi\hbar^4}{m^2\mathcal{R}(\mathbf{k} = \mathbf{0})} \gg \frac{\hbar}{\sqrt{2mgn}}, \tag{33}$$

which means that the Larkin length $\ell_{\mathrm{L}} = \pi^{3/2}\hbar^4/\left[m^2\mathcal{R}(\mathbf{k} = \mathbf{0})\right]$ [47, 48], associated with the pinning energy due to the disorder, is much larger than the healing length $\xi$.

## 4 Condensates in disordered traps

Recently, it has been demonstrated that the Huang-Meng theory can be verified quantitatively by using the homogeneous results within a local density approximation (LDA) [33]. Here we demonstrate that the above time-dependent results can also be extended to the case of inhomogeneous condensates in disordered traps, provided that the disorder correlation length is much smaller than the spatial extent of the atomic cloud. Thus, we assume that in addition to the random potential $u(\mathbf{x}, t)$ a trapping potential $U(\mathbf{x})$ is applied. In the absence of disorder, the real-valued ground state wave function $\Psi_0(\mathbf{x})$ satisfies

$$\left(-\frac{\hbar^2\nabla^2}{2m} + U(\mathbf{x}) - \mu_0 + g\Psi_0^2(\mathbf{x})\right)\Psi_0(\mathbf{x}) = 0, \tag{34}$$

which leads to the following expression for the chemical potential

$$\mu_0 = -\frac{\hbar^2}{2m}\frac{\nabla^2\Psi_0(\mathbf{x})}{\Psi_0(\mathbf{x})} + U(\mathbf{x}) + g\Psi_0^2(\mathbf{x}). \tag{35}$$

Upon switching the disorder on at $t = 0$, the system dynamics is governed by

$$i\hbar\frac{\partial}{\partial t}\Psi(\mathbf{x}, t) = \left(-\frac{\hbar^2\nabla^2}{2m} + U(\mathbf{x}) + u(\mathbf{x})f(t) - \mu_0 + g|\Psi(\mathbf{x}, t)|^2\right)\Psi(\mathbf{x}, t), \tag{36}$$

where we explicitly accounted for the initial phase evolution $e^{-i\mu_0 t/\hbar}$. As in the homogeneous case, we are going to invoke the perturbative expansion (6) for the wave function $\Psi(\mathbf{x}, t)$. In the first order, the last equation yields

$$i\hbar\frac{\partial}{\partial t}\Psi_1(\mathbf{x}, t) = \left(-\frac{\hbar^2\nabla^2}{2m} + \frac{\hbar^2}{2m}\frac{\nabla^2\Psi_0(\mathbf{x})}{\Psi_0(\mathbf{x})} + g\Psi_0^2(\mathbf{x})\right)\Psi_1(\mathbf{x}, t)$$
$$+ g\Psi_0^2(\mathbf{x})\Psi_1^*(\mathbf{x}, t) + \Psi_0(\mathbf{x})u(\mathbf{x})f(t), \qquad (37a)$$

$$-i\hbar\frac{\partial}{\partial t}\Psi_1^*(\mathbf{x}, t) = \left(-\frac{\hbar^2\nabla^2}{2m} + \frac{\hbar^2}{2m}\frac{\nabla^2\Psi_0(\mathbf{x})}{\Psi_0(\mathbf{x})} + g\Psi_0^2(\mathbf{x})\right)\Psi_1^*(\mathbf{x}, t)$$
$$+ g\Psi_0^2(\mathbf{x})\Psi_1(\mathbf{x}, t) + \Psi_0(\mathbf{x})u(\mathbf{x})f(t), \qquad (37b)$$

where we eliminated the chemical potential using (35). Averaging (37) over disorder realizations leads to the relations

$$i\hbar\frac{\partial}{\partial t}\langle\Psi_1(\mathbf{x}, t)\rangle = \left(-\frac{\hbar^2\nabla^2}{2m} + \frac{\hbar^2}{2m}\frac{\nabla^2\Psi_0(\mathbf{x})}{\Psi_0(\mathbf{x})} + g\Psi_0^2(\mathbf{x})\right)\langle\Psi_1(\mathbf{x}, t)\rangle + g\Psi_0^2(\mathbf{x})\langle\Psi_1^*(\mathbf{x}, t)\rangle, \quad (38a)$$

$$-i\hbar\frac{\partial}{\partial t}\langle\Psi_1^*(\mathbf{x}, t)\rangle = \left(-\frac{\hbar^2\nabla^2}{2m} + \frac{\hbar^2}{2m}\frac{\nabla^2\Psi_0(\mathbf{x})}{\Psi_0(\mathbf{x})} + g\Psi_0^2(\mathbf{x})\right)\langle\Psi_1^*(\mathbf{x}, t)\rangle + g\Psi_0^2(\mathbf{x})\langle\Psi_1(\mathbf{x}, t)\rangle, \quad (38b)$$

which reveal the important fact that $\langle\Psi_1(\mathbf{x}, t)\rangle$ and $\langle\Psi_1^*(\mathbf{x}, t)\rangle$ are not at all influenced by the disorder. Moreover, due to the initial conditions $\Psi_1^{(*)}(\mathbf{x}, t = 0) = 0$, we conclude that $\langle\Psi_1^{(*)}(\mathbf{x}, t)\rangle = 0$ holds at all times, as in the homogeneous case.

Since, by our presumption, the disorder correlation length $\sigma$ is much smaller than the characteristic condensate width $L_0$, the perturbative corrections of the condensate wave function $\Psi_1(\mathbf{x}, t)$ and $\Psi_1^*(\mathbf{x}, t)$ vary in space much more rapidly than $\Psi_0(\mathbf{x})$. Hence, in order to determine them from (37) we employ LDA, in the sense of [33], by treating $\Psi_0(\mathbf{x})$ as a constant background quantity $\Psi_0$ that will, at the end of the calculation, be restored to its original value at the point $\mathbf{x}$. In the same spirit, we can make the estimates

$$\left|\frac{\nabla^2\Psi_0(\mathbf{x})}{\Psi_0(\mathbf{x})}\Psi_1^{(*)}(\mathbf{x}, t)\right| \sim \frac{\left|\Psi_1^{(*)}(\mathbf{x}, t)\right|}{L_0^2} \ll \left|\nabla^2\Psi_1^{(*)}(\mathbf{x}, t)\right|, \qquad (39)$$

so that under LDA the equations (37) become in fact identical to their homogeneous counterparts (11). In that way all the results from the previous sections can effortlessly be generalized to the considered trapped case by making simple substitutions $\Psi_0 \mapsto \Psi_0(\mathbf{x})$ and $n \mapsto \Psi_0^2(\mathbf{x})$ therein.

Let us now illustrate the above findings in the strongly interacting Thomas-Fermi limit. In such a case the weak disorder criterion (33) is more likely to hold. We assume that the disorder is a delta-correlated one (25). The initial wave function takes the form

$$\Psi_0^{\text{TF}}(\mathbf{x}) = \sqrt{n_0}\left(1 - \frac{x_1^2}{\varrho_1^2} - \frac{x_2^2}{\varrho_2^2} - \frac{x_3^2}{\varrho_3^2}\right)^{1/2}, \qquad (40)$$

whenever $x_1^2/\varrho_1^2 + x_2^2/\varrho_2^2 + x_3^2/\varrho_3^2 \leq 1$ and is equal to zero otherwise. The total particle number is given by

$$N = \frac{2}{5}n_0 V, \qquad (41)$$

while $V = 4\pi\varrho_1\varrho_2\varrho_3/3$ is the volume of the ellipsoid characterized by the Thomas-Fermi radii $\varrho_1$, $\varrho_2$, $\varrho_3$. Obviously, the peak density $n_0$ is 5/2 larger than the average density $N/V$.

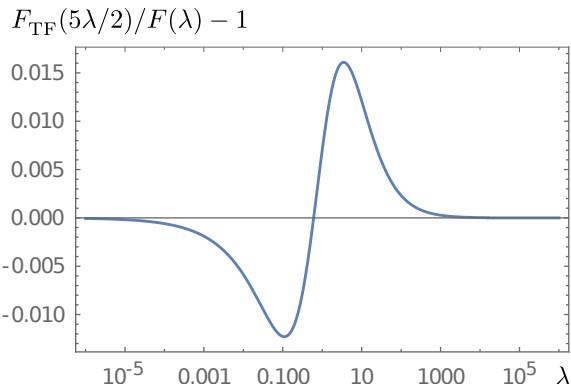

Figure 5: Relative deviation between the Thomas-Fermi form-function (44) and its homogeneous counterpart (28).

After applying the LDA, the stationary condensate deformation (27) becomes spatially dependent $q_\tau^{\mathrm{TF}}(\mathbf{x})$. Integrating it over the Thomas-Fermi ellipsoid leads to the total stationary condensate deformation

$$Q_\tau^{\mathrm{TF}} = \int_V d^3x \, q_\tau^{\mathrm{TF}}(\mathbf{x}) = F_{\mathrm{TF}}(g n_0 \tau/\hbar) Q_{\mathrm{HM}}^{\mathrm{TF}}, \tag{42}$$

where the total equilibrium condensate deformation reads

$$Q_{\mathrm{HM}}^{\mathrm{TF}} = R \frac{3m^{3/2}V}{64\hbar^3\sqrt{g}}\sqrt{n_0}. \tag{43}$$

The switch on time $\tau$ appears as an argument of the form-function

$$F_{\mathrm{TF}}(\lambda) = \frac{6(5\lambda^4 - 6\lambda^2 + 8\lambda + 9)\arctan(\sqrt{\lambda}) - \sqrt{\lambda}\{5\lambda[-6\lambda^2 + 3\pi(\lambda^2 - 2)\sqrt{\lambda} + 2\lambda + 6] + 54\}}{12\pi\lambda^2}, \tag{44}$$

which is a generalization of (28) to the Thomas-Fermi case, with the same limiting behavior $\lim_{\lambda\to\infty} F_{\mathrm{TF}}(\lambda) = 1$ and $\lim_{\lambda\to 0} F_{\mathrm{TF}}(\lambda) = 5/2$. Thus, akin to the homogeneous result (29), we conclude that

$$\lim_{\tau\to\infty} Q_\tau^{\mathrm{TF}} = Q_{\mathrm{HM}}^{\mathrm{TF}}, \qquad \lim_{\tau\to 0} Q_\tau^{\mathrm{TF}} = \frac{5}{2} Q_{\mathrm{HM}}^{\mathrm{TF}}. \tag{45}$$

In order to make a closer comparison with the corresponding homogeneous situation, we choose the average density $N/V$ to match the homogeneous density $n$. According to (41), this yields $n_0 = 5n/2$. Hence, in Fig. 5 we plot the relative difference $F_{\mathrm{TF}}(5\lambda/2)/F(\lambda) - 1$. Remarkably, the relative deviation between the two form-functions is below 1.7% across the entire domain. This indicates that a simple homogeneous analysis can be quite valuable in reliably estimating the overall effect of weak disorder also in the inhomogeneous trapped settings, provided that the appropriate quantities are properly matched.

## 5 Conclusion and outlook

Our results clearly demonstrate that the condensate deformation is an indicator of the non-equilibrium feature of steady states of a Bose gas in a temporally controlled weak disorder. However, the exact nature of the observed steady states after switching the disorder on, and

potentially off afterwards, still remains an elusive point, which warrants further investigation. Additional understanding could be gained, for instance, by proceeding along the route of Ref. [38]. Namely, it is a natural follow-up question to ask how much of the superfluid density would survive in the switching on and off scenarios considered here. Indeed, previous studies showed that superfluidity is more affected by disorder than the condensate itself both in equilibrium [19–22, 24, 49, 50] and in non-equilibrium [38]. Additional characterization of the emerging steady states would be gained by examining the time scales necessary to reach the respective stationary values of the condensate deformation and the superfluid fraction, in particular how they depend on the used protocols.

Our findings potentially have broader implications. The equilibrium and the dynamical contributions to the induced condensate deformation suggest that the same could generically hold for other physical quantities of interest, at least in the perturbative weak disorder regime. Moreover, it is both an interesting and intriguing question how the dynamical condensate deformation would behave in the non-perturbative strong disorder setting [47, 48, 50–52], where the two aforementioned contributions might not be additive anymore. As this problem is quite analytically challenging, we hope that numerical simulations and experiments could shed the first light upon that situation.

Furthermore, the present work can be extended in multiple directions as the disorder driving could be of different types. One can, for instance, envision varying temporally some other feature of the disorder, such as its correlation length, instead of its strength. Alternatively, time-periodic driving has a great potential to expand the scope of the achievable non-equilibrium asymptotic states [53–57]. Also, there is an emerging and quite promising research direction of utilizing stochastic driving with a tunable time scale [58]. Moreover, one can ask, in the spirit of quantum control theory [59–61], whether a particular driving protocol could be designed in order to maximize or minimize the final condensate deformation.

## Acknowledgments

We thank Antun Balaž, Benjamin Nagler, and Artur Widera for insightful discussions.

**Funding information** M. R. and A. P. acknowledge financial support by the Deutsche Forschungsgemeinschaft (DFG, German Research Foundation) via the Collaborative Research Center SFB/TR185 (Project No. 277625399) and via the Research Unit FOR 2247 (Project No. PE 530/6-1).

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
