# Peer review of "Non-equilibrium evolution of Bose-Einstein condensate deformation in temporally controlled weak disorder"

_SciPost Physics, doi:SciPost Phys. 10, 008 (2021)_

## Round 1 · Referee Report · Anonymous (Referee 2) · 2020-10-26

Report

Dear Editor,

the manuscript "Non-equilibrium evolution of Bose-Einstein condensate
deformation in temporally controlled weak disorder" by Radonjic and Pelster
reports an interesting calculation of the condensation deformation emerging from the switch-on (or -off) of an external disordered potential.

Remarkably, they significantly generalize the seminal calculation by Huang and Meng (Ref. [19] ) by going beyond the adiabatic limit. Indeed, they report that the condensate deformation is actually defined by two contribution, one of them explicitly dependent on the switch-on/off protocol. This dynamical correction signals that disorder can drive the system towards a stationary out-of-equilibrium states.

In my opinion, this manuscript surely deserves publication in SciPost: the object of the theoretical investigation is timely and clear, while the analytical reasoning are clearly exposed and seemingly easy to reproduce and generalize.

Before final acceptance, I have only few minor remarks, just to improve the readability for a wider audience:

1) The author properly spend a certain amount of words in outlining the experimental possibilities to engineer a disorder landscape for the condensate. It may be also worth to mention how the switch-on/off protocols can be implemented in an actual cold-atom experiment, if there are limitations or technical obstacles to overcome...

2) Eq. (10) is a perturbative approximation up to the second order. Have the authors approached in any way the calculation of the following terms? Do they result in simple shift of the computed profile (for instance in Fig. 1), or there may be dangerous behaviours of the perturbative series?

3) It can be useful to have some conditions stating when a disorder potential is considered "weak", which seems a relevant point since we are actually within a perturbative framework.

  • validity: high
  • significance: high
  • originality: top
  • clarity: top
  • formatting: excellent
  • grammar: good

Author:  Milan Radonjic  on 2020-11-30  [id 1067]

(in reply to Report 2 on 2020-10-26)
Category:
answer to question

We are grateful to the Referee for recognizing the value of our work and suggesting its publication in the SciPost journal. We are also thankful for the given remarks, which refined our paper and made it more accessible to a wider audience. Below we address the specific changes, in a point-by-point manner:

1) In section 3, immediately after the equation (24), we mention potential technical and physical constraints concerning the disorder switch on time in the experimentally feasible cold-atom setups.

2) Higher than the second order terms for q(t) would, of course, assume the knowledge of \Psi_2(x,t), \Psi_3(x,t), etc. More concretely, the third order correction for q(t) involves \Psi_2(x,t) and \Psi_2^*(x,t). The perturbative procedure yields the equations for \Psi_2(x,t) and \Psi_2^*(x,t) of the same form as (11), but with the source terms of the second order which involve u(x) and \Psi_1(x,t). The corresponding solutions would contain nested temporal convolutions and, thus, would not lead to any practical analytic final result. Therefore, we did not pursue their definitive evaluation. However, one can state with certainty that higher than second order correlation functions of the disorder would become important. Based on the finite second order corrections to the equilibrium wave function available in the literature, we expect that at least the third order correction for q(t) would be finite.

3) This important point is discussed at the end of section 3.3, where it is shown that for the disorder to be weak it is necessary that its Larkin length is much larger than the condensate healing length.

---

## Round 1 · Referee Report · Anonymous (Referee 1) · 2020-10-26

Strengths

1) The authors discuss a general approach for dealing with the effects of switching on/off a disordered potential on the density deformation of a Bose-Einstein condensate.

2) The paper is written very clearly, and meets all the General acceptance criteria of this Journal.

3) It proposes a new pathway in this research field.

Weaknesses

1) It is limited to the study of homogeneous systems, the effect of trapping potentials are only mentioned in the conclusions at a qualitative level, very briefly.

2) It would have been very interesting to compare the analytical results with the numerical solution of the Gross-Pitaevskii equation. This should be relatively easy to do.

Report

This is a very interesting paper, very clearly written and presented. It represents a timely contribution to the current investigation of ultracold atoms in the presence of disorder. The approach and the results discussed here are likely to trigger further studies along the direction outlined in the paper. Apart from minor suggestion I list below, I would recommend the author to consider comparing their nice analytical results with Gross-Pitaevskii (GP) simulations. This would be the natural completion of this work because it would allow for a verification of the perturbative analytical estimates against the exact behavior of the system (here, the GP equation). With this extension, I would be pleased to recommend the manuscript for publication on SciPost Physics. Otherwise, I reckon this paper more suited for the Physics Core journal.

Requested changes

1) It would be helpful to specify (possibly already in the abstract) that the "equilibrium part" of the the stationary condensate deformation corresponds to adiabatic switching on the disorder potential.

2) The wording "in the frame rotating with the frequency μ0/􏰓hbar" may be confusing, there are no _rotating_ frames.

3) I recommend to extend the discussion on how the presence of a trap would affect the present results.

4) A comparison with the results of GP simulations would be very useful.

  • validity: high
  • significance: good
  • originality: high
  • clarity: top
  • formatting: excellent
  • grammar: excellent

Author:  Milan Radonjic  on 2020-11-30  [id 1066]

(in reply to Report 1 on 2020-10-26)
Category:
answer to question

We thank the Referee for a positive report, appreciation of our work and useful suggestions that considerably improved our manuscript and extended its range of applicability. Below we make a point-by-point reply to the specific Referee’s comments.

1) The correspondence of the equilibrium part to adiabatic switching on is now explicitly stated in the abstract.

2) We have now used the more appropriate wording: “after explicitly separating out the initial phase evolution exp(-i\mu_0 t/\hbar).”

3) The trapped case is now discussed in the new section 4. Therein, we demonstrate that the homogeneous time-dependent results can readily be extended to the case of inhomogeneous condensates in disordered traps, provided that the disorder correlation length is much smaller than the spatial extent of the atomic cloud. The backbone of such an extension is the local density approximation discussed in our previous publication New J. Phys. 22, 033021 (2020). A new figure 5 is added to stress that the homogeneous analysis can be quite valuable in reliably estimating the overall effect of weak disorder in the inhomogeneous trapped settings, provided that the appropriate quantities are properly matched.

4) We definitely agree with the Referee that a comparison with the Gross-Pitaevskii (GP) simulations would be very useful. Since there are nowadays freely available numerical codes for that purpose, it could be relatively easy to do so, at least from the coding side. However, here we would need full-fledged 3D time-dependent simulations, with the addition of the statistical sampling of the disorder distribution. According to Figure 1, the time propagation until stationarity in a single disorder realization would have to be many orders of magnitude larger than the characteristic mean-field time scale. Thus, the complete comparison would require considerable time and extensive computational resources. Moreover, the full GP simulations would open the door into the strong disorder regime, which is beyond the scope of the present paper. Therefore, we choose to leave this for future work.

In line with the aforementioned changes, we have adapted the text of the manuscript for consistency and readability. Also, we have added 4 new references [47], [48], [51], [52].

---

## Round 2 · Referee Report · Anonymous (Referee 3) · 2020-12-18

Report

While my previous report was already in support of publishing this interesting manuscript, I am satisfied with the authors' reply to my comments.

I think that this revised version is even slightly improved and I certainly recommend its acceptance.

---

## Round 2 · Referee Report · Anonymous (Referee 4) · 2020-12-22

Report

With this revised manuscript the authors have satisfactorily addressed most of the remarks of both Referees, and they have modified the manuscript accordingly. The only point that has been left out is the comparison with the numerical solution of the Gross-Pitevskii (GP) equation, that the authors estimate to require considerable time and extensive computational resources. I do not disagree with the authors on this, but I think that contrasting the approximate analytical approach with the exact (though numerical) GP solution would be necessary to justify publication in the flagship SciPost Physics journal. In any case, I confirm that this is a very interesting paper, very clearly written and presented, and I am pleased to recommend it for publication in SciPost Physics Core.

---

## Editorial Decision

published